# The Complement System, Aging, and Aging-Related Diseases

**DOI:** 10.3390/ijms23158689

**Published:** 2022-08-04

**Authors:** Runzi Zheng, Yanghuan Zhang, Ke Zhang, Yang Yuan, Shuting Jia, Jing Liu

**Affiliations:** Laboratory of Molecular Genetics of Aging and Tumor, Medical School, Kunming University of Science and Technology, 727 South Jing Ming Road, Kunming 650500, China

**Keywords:** complement system, C3, aging

## Abstract

The complement system is a part of the immune system and consists of multiple complement components with biological functions such as defense against pathogens and immunomodulation. The complement system has three activation pathways: the classical pathway, the lectin pathway, and the alternative pathway. Increasing evidence indicates that the complement system plays a role in aging. Complement plays a role in inflammatory processes, metabolism, apoptosis, mitochondrial function, and Wnt signaling pathways. In addition, the complement system plays a significant role in aging-related diseases, including Alzheimer’s disease, age-related macular degeneration, and osteoarthritis. However, the effect of complement on aging and aging-related diseases is still unclear. Thus, a better understanding of the potential relationship between complement, aging, and aging-related diseases will provide molecular targets for treating aging, while focusing on the balance of complement in during treatment. Inhibition of a single component does not result in a good outcome. In this review, we discussed the research progress and effects of complement in aging and aging-related diseases.

## 1. Introduction

The complement system plays a central role in innate immunity. According to the “body fluid theory”, the presence of antiviral and bactericidal substances in bodily fluids is dependent on the presence of two factors. Heat-stable factors include antibodies, while a heat-unstable factor is the complement system [1]. Similar to antibodies, complement plays a crucial role in resisting foreign substances in the immune system. During the recognition process, the complement system produces a membrane attack complex (MAC) to detect and remove foreign pathogens. As scientific research has expanded, we have gradually learned that the complement system is closely linked to a number of diseases, including myocardial infarction [2], systemic lupus erythematosus [3], and metabolic syndrome [4].

Currently, population aging is a major public health concern worldwide. There were over 962 million people over the age of 65 in 2017, and this population is growing at a 3% annual rate [5]. Aging itself is a complex and chronic process. In addition, the World Health Organization (WHO) has determined that aging can be treated as a disease under the International Classification of Diseases (ICD-11) [6]. In this “diseased” state, the structures and functions of the cells degrade. The body’s lack of balance eventually leads to a variety of ailments, such as chronic illnesses. People have begun to rely not only on chronological age (CA) to describe aging, but also on biological age (BA) to define aging. According to research, the complement components C3 and C4 are favorably connected with metabolic syndrome and inversely correlated with life expectancy in centenarians [7]. C3 and C4 are associated with daily physical activities in centenarians [4], and abdominal obesity in centenarians was strongly correlated with complement C3 levels [8]. This finding suggests that the complement system and aging are linked through some pathways. The association between complement and aging, on the other hand, is less well understood, and the processes are unclear. Therefore, this review summarizes the relationship between complement and aging and considers whether complement could be a potential target for the treatment of aging-related diseases, focusing on the balance of complement in the organism during treatment.

## 2. The Complement System

### 2.1. Composition of the Complement System

Complement can be traced back 350 million years to horseshoe crabs. In 1891, Buchner and colleagues discovered that blood was resistant to bacterial heat destabilizers, and in 1899, Paul Ehrlich used the term complement to describe the antibacterial and heat-sensitive compounds found in blood [1,3]. The complement system is composed of more than 40 different proteins, the majority of which are plasma proteins or membrane proteins produced by the liver, as well as other proteins that work together. Complement components were originally named in the order in which they were identified: C1, C2, and C3. Until now, complement components have been named following the order of activation: C1, C4, C2, C3, C5, C6, C7, C8, and C9 [1,9,10].

C1 is a critical component of the initiation of the classical pathway (CP), which is a complex made up of C1q, C1r, and C1s that separates after C1 is activated. C1q has a molecular weight of 400 kDa and can bind and recognize proteins on bacteria and viruses. C1r is a β-globulin that activates C1s by breaking the thioester connection between them. C1s is an alpha globulin and a serine protease that cleaves C4 and C2 after C1r activation [11].

C4 and C2 are also critical components of the complement system, and they play key roles in activating the complement cascade in both the CP and lectin pathway (LP). C4 is a disulfide-bonded three-chain glycoprotein composed of a-chain, b-chain, and g-chain. When the complement system is activated, activated C1s cleave C4 into two parts, C4a and C4b, and the generated C4b binds C2a (small fragment generated by C1s cleavage of C2) to form C4b2a (a C3 convertase). C4b can also cleave C4d (an inactive split product), and its accumulation can be used as a marker of complement activation [12,13]. C4a is an allergenic toxin [14].

C3 is the central point of the three-cascade activation pathway of the complement system and is composed of alpha and beta chains. C3 converting enzyme cleaves C3 into C3a small fragments and C3b large fragments in vivo, and the presence of C3b aids in the formation of C5 converting enzyme, while C3b also produces C3d, which acts as a receptor that inhibits complement factor I [15]. C3a is also an allergenic toxin [16].

C5 is the MAC component carrier. C5 convertase cleaves C5 into two parts: C5a and C5b. C5a is the complement system’s most potent allergenic toxin, and C5b is a hydrophobic molecule that serves as the precursor to MAC formation [11].

### 2.2. Activation of the Complement System

Classical pathway, lectin pathway, and alternate pathway are three commonly accepted pathways of complement system activation. The main difference between the three pathways is the initiation process. After initiation, C3 convertase is formed and further cleaves C3. The C3b fragment binds to the preceding complex to form C5 convertase which cleaves C5b and C5a, causing the assembly of the MAC, which lyses cells and stimulates the production of an inflammatory response to remove foreign substances.

#### 2.2.1. Classical Pathway

The classical pathway (CP) is often initiated by C1q attaching to an immunological complex (e.g., lgM or lgG). C1q then activates C1r, which changes the conformation of the C1r_2_—C1s_2_ structure. C1r liberates C1s, and C1s has serine protease (SP) activity and cleaves C4 and C2 to form C4b2a (C3 convertase). C3 convertase forms C5 convertase by cleaving C3 and later binding to C3b. C4a, C3a, and C5a stimulate inflammatory responses. C5b forms the C5b-9 MAC with the help of C6, C7, C8, and C9 to attack host cells or pathogens [10].

#### 2.2.2. Lectin Pathway

The activation of the lectin pathway (LP) is identical to that of the CP, except that the initiator of LP is mannose-binding lectin (MBL), which forms multimeric lectin complexes by binding with ficolin. This binding causes the activation of MBL-associated serine protease (MASP), which triggers the complement system. MASP-1 and MASP-2 are similar to C1r and C1s, respectively. MASP-1 and MASP-2 fully activate the complement system by cleaving C4 and C2 to form C3 convertase.

#### 2.2.3. Alternative Pathway

The alternative pathway (AP) normally acts as a “monitor”, and C3(H_2_O) (Hydrolysis of C3) works slowly and at low levels. This factor is produced by the spontaneous hydrolysis of the unstable C3 thioester bond and the conversion of C3 to the active C3(H_2_O) form without activating the complement pathway. This factor is kept at a low level to monitor pathogenic invasion. A substantial number of C3 thioester structural domain (TED) undergo metastability when pathogen invasion is detected, exposing C3 to the factor b (Fb) binding site. C3(H_2_O) binds to Fb and is cleaved by the serine protease factor D (Fd) to form a C3 convertase (C3bBb) that is distinct from CP and LP. C3 convertase breaks down C3 into C3a and C3b, and C3b acts as an active fragment that binds to pathogens or target cells. In response to C3 convertase, C3b generates C5 convertase, which cleaves C5 to recruit a membrane attack complex (MAC) to lyse pathogens (Figure 1).

Additionally, C3b and C3(H_2_O) can cause local AP activation by being attracted to the cell surface by Properdin (FP), which improves C3b and Bb binding and increases the stability of C3 convertase [10]. Other researchers have recently postulated that in the absence of C3, thrombin can operate as a complement activator instead of C5 convertase, thus suggesting a novel mechanism of complement activation [1,10,17,18].

The primary function of the complement system is to protect against foreign substances by recruiting immune cells to phagocytose or lyse target cells. However, complement activation is essential for the complement system to function, and C3 and C3 convertase play vital roles in this process. C3 convertase cleaves C3 into C3a and C3b, which are involved in a complement-activated autoreactivation cycle via AP. Then, as a large amount of C3a and C5a can be produced to recruit immune cells, C3b binds to C3 convertase to create C5 convertase, and C5b and other components eventually combine to form a large number of MACs to play a protective role [1]. Thus, C3 plays a large role in facilitating the activation of complement.

## 3. Complement and Aging

As human beings live longer, they react to external stresses and accumulate harmful substances in the body, and eventually the physiological integrity of tissues and organs is gradually lost and their functions are impaired, which is a process we call aging [19]. Common hallmarks of aging include genomic instability, telomere attrition, epigenetic alterations, the loss of proteostasis, mitochondrial dysfunction, cellular senescence, and other underlying mechanisms [20,21]. The complexity of the phenomenon makes aging a difficult process to understand, but inflammation is the key to abnormalities associated with the aging process [22,23,24]. This slow and widespread inflammatory process is sensed by C3(H_2_O) in the AP, which activates the complement system. Elevated complement levels can act as immunomodulatory agents and clear the accumulation of harmful substances from the organism. Complement C3 and C4 levels have been shown to correlate with age [5], and longevity has been shown to have a negative connection with C3/C4 levels, suggesting that high C3 levels are harmful to longevity. Another study demonstrated that low levels of C3 delayed renal senescence [25] and that the use of Radix polygalae saponins to affect C3 expression could also extend the lifespan of *C. elegans* [26]. It is clear that complement and aging are closely linked, but the relationship between them is unclear. Complement affects inflammation, metabolism, apoptosis, mitochondrial function, and Wnt signaling pathway. Complement may act through these components in the biological process of organismal aging.

### 3.1. Complement and Inflammation

C3a is the product of the cleavage of C3 by C3 convertase, and is an allergenic toxin that can mediate inflammation. C4a and C5a have similar functions [27]. In guinea pig macrophages [28], C3a could stimulate cellular responses such as the Ca^2+^ response and O_2_ release, while in human eosinophils, C3a is an effective activator of transitory Ca^2+^ alterations and reactive oxygen species formation [29]. Different concentrations of C3a can induce ROS production by human polymorphonuclear neutrophils (PMNs), and C3a is a potential activator of the PMNs respiratory burst [30]. Reactive oxygen species (ROS) mediate the activation of NLRP3 inflammatory vesicles, and the accumulation of ROS is an important cause of pathological aging [31,32]. The C3a/C3aR signaling axis can induce IL-1β secretion from monocytes by enhancing ATP efflux and activating the NLRP3 inflammatory vesicle through extracellular signal-regulated protein 1/2 (ERK1/2) [33]. TNF-α and interleukin IL-1β gene expression and protein synthesis in human peripheral blood mononuclear cells is also affected by C3a and C3a desArg (C3a lacking C-terminal arginine) [34]. Therefore, we know that C3a has a proinflammatory effect, but interestingly, C3a has an anti-inflammatory effect under different conditions. C3a can bind C3a receptor (C3aR) in acute intestinal injury to reduce neutrophil mobilization and ameliorate intestinal ischemia-reperfusion pathology in mice [35]. As a result, the role of C3 in inflammation is more complicated than simple magnification.

### 3.2. Complement and Metabolism

Complement C3 is strongly associated with lipid metabolism, cardiovascular disease, metabolic syndrome, and diabetes [4,36]. C3 is largely generated in the liver [37], and studies have shown a substantial linear association between C3 levels and blood lipids, waist circumference, and C-reactive protein (CRP) [38]. However, other tissues of the body, such as adipose tissue, can create C3, and age-related showing of the metabolism combined with the accumulation of calories can easily lead to obesity [39], which could explain why C3 levels increase. Complement C3 affects both steatosis and the inflammatory response in the liver. Even with 60% liver fat, geese do not show inflammatory or pathogenic features, due to C3 representative downstream genes of C3 (FASN and ETNK1) that regulate C3 function [40]. In addition to obesity, a study of 2815 nondiabetic healthy middle-aged men with long follow-up plasma protein analysis showed that complement C3 levels were associated with the risk of developing diabetes [41]. C3 can also be used as a predictor of cardiovascular or coronary events [42]. Complement protein deposition and activation in the arterial wall can be observed in patients with coronary artery disease. Activation of the complement system is hypothesized to play a role in acute coronary crises. In ruptured and fragile plaques, deposits of iC3b (inactivated C3b) are increased, and when the plaque ruptures, the lesion components are discharged into the bloodstream, further activating the complement system in the artery [43].

### 3.3. Complement and Apoptosis

Apoptosis is a self-directed and orderly form of cell death that is under genetic control, and it is a means to eliminate unwanted cells without triggering an inflammatory response. The activation of apoptotic mechanisms removes senescent cells and allows the body to function normally. Senescent cells are resistant to apoptosis and require assistance to further induce apoptosis [44]. C3 can lead to apoptosis by forming the MAC via the complement-activated pathway. The complement system (which generates significant levels of C3a/C5a/MAC) can be stimulated by lipopolysaccharide (LPS) and cobra venom factor (CVF) to trigger endothelial cell death. In a glaucoma mouse model, the MAC can also trigger apoptosis, which results in the death of retinal ganglion cells [45,46,47]. iC3b, an inactivated form of C3b, can also promote apoptosis through immunomodulatory effects [47]. However, C3 tends to have diverse effects depending on the situation. When C3a was cultured with human macrophages, the macrophage apoptotic rate decreased with increasing complement C3a concentrations, according to one study [48]. Complement C3 can also prevent imiquimod-induced bullous-like skin inflammation by inhibiting apoptosis [49]. Moreover, C3 interacts with the autophagy-associated protein ATG16L1 to regulate autophagy and protect β-cells from cytokine-induced apoptosis [50]. Thus, islet β-cell apoptosis can be increased by C3 inhibition [51]. The amount of complement molecules in our bodies is crucial, and if we want to use these molecules for our benefit, we must first identify a proper state, because complement has varying effects depending on the conditions.

### 3.4. Complement and Mitochondrial Function

Damage to DNA and existing proteins arises from mitochondrial metabolism impairment. Such deleterious effects are part and parcel of the aging process [52]. C1qbp (C1q binding protein) is a mitochondrial protein. A study suggests a specific function of C1qbp in the brain related to mitochondria, such as the regulation of local energy supply in neuronal cells [53]. There is an indication of lectin pathway involvement in mitochondrial immune handling [54]. However, mitochondria are destroyed by complement. Human renal proximal tubular epithelial cells exposed to C5a in vitro produced reactive oxygen species and had impaired mitochondrial respiratory activity. When the lipidomics signature was examined, it was discovered that the diabetic kidney had aberrant cardiolipin remodeling, which is a cardinal sign of dysfunctional mitochondrial architecture and bioenergetics. An oral C5aR1 inhibitor (PMX53) delivered in vivo reversed the phenotypic alterations and restored the renal mitochondrial fatty acid composition [55]. PC12 cells exposure to C5a led to inhibition of mitochondrial respiration, dehydrogenase, and cytochrome c oxidase activities [56]. Treatment with H_2_O_2_ induced a dose-dependent increase in C3a-mtC3aR colocalization. C3a increased mitochondrial Ca_2_^+^ uptake in isolated mitochondria from H_2_O_2_-treated cells, which could be inhibited by C3aR antagonism (SB290157), mitochondrial Ca_2_^+^ uniporter blocker (Ru360), and G-protein inhibition (pertussis toxin, PTX). C3a also inhibited mitochondrial respiration in a way that was dependent on both SB290157 and PTX. Oxidative stress increases mtC3aR, leading to altered mitochondrial calcium uptake and ATP production [57]. Perhaps at first complement is beneficial for mitochondria, but increasing age, abnormal increase, or activation of complement leads to mitochondrial dysfunction, thus affecting aging.

### 3.5. Complement and Wnt Signaling Pathway

Wnts are a wide family of secreted proteins that influence a variety of cellular responses during development and cause intracellular signaling that is evolutionarily conserved. Studies have shown that Wnt signaling is involved in the control of mammalian aging [58]. Wnt/β-catenin signaling is augmented in a mouse model of accelerated aging, and suppression of canonical the aging-related decrease of skeletal muscle regeneration is reversed by Wnt signaling. Moreover, C5a stimulation of RTEC (Renal tubular epithelial cell) led to up-regulation of SA-β Gal and cell cycle arrest markers such as p53 and p21. Accordingly, they discovered that elevated Wnt4 and catenin expression correlated with SA-Gal, p21, p16, and IL-6 positivity in a swine model of renal I/R damage. Complement Inhibitor (C1-Inh) administration inhibited Wnt4/catenin activation, decreased SA-Gal, p21, p16, and IL-6 levels, and blocked the senescence-associated secretory phenotype (SASP) [59]. Moreover, aging-related increases in C1q secretion cause muscular atrophy and fibrosis. The study hypothesizes that muscle Wnt signaling in senescent mice, which is increased by resistance training, may help to avoid muscle fibrosis and atrophy [60]. The canonical Wnt/β-catenin signaling pathway was activated in lupus nephritis patients [61]. The pathogenesis of lupus nephritis is closely related to complement [62]. C5b-9 serum significantly activated the Wnt/β-catenin signaling pathway and promoted autophagy [63]. The above could indicate that complement is closely related to Wnt and that complement may act through this pathway in aging.

## 4. Complement and Aging-Related Diseases

Aging is one of the major risk factors for the development of diseases, such as metabolic diseases [64], neurodegenerative diseases [65], and cardiovascular diseases [66], in the human body. Although the complement system affects the body through immune regulation, dysregulation of complement in aging-related diseases is more likely to accompany disease and exacerbate disease onset.

### 4.1. Alzheimer’s Disease

Alzheimer’s disease (AD) is a neurological illness that affects elderly individuals. Amyloid plaques and neurofibrillary tangles (NFTs) are extracellular deposits of amyloid β protein (Aβ) that characterize this disease. Neuronal fiber tangles are composed of paired helical filament (PHF) aggregates of hyperphosphorylated tau proteins [67,68]. As early as the last century, studies have shown elevated levels of C1q, C3, and C4 colocalized with Aβ plaques in the brain tissue of AD patients and elevated levels of C3 and C4 mRNA in the temporal lobe of the brain [69,70]. In 2018 an investigator found high complement levels in astrocytes isolated from the brains of AD patients [71]. C1q can bind to neurons that express calreticulin (CRT) and cause neuronal ROS production and damage, whereas the presence of the MAC may also contribute to neuronal loss and degeneration in AD [68,72]. Mice lacking C1q or C3 have persistent synapse elimination abnormalities in the CNS [73]. Crossing a C1q-deficient (APPQ^−/−^) mouse model with an amyloidosis (APPPS1Q^−/−^) mouse model resulted in a significant reduction in glial cell activation compared to amyloidosis (APPPS1Q^−/−^) mice, which has been linked to a variety of neurological diseases [74]. C3 knockout in a mouse model with tau protein lesions (TauP301S) improved neuronal loss and brain atrophy in mice [75]. An anti-pGlu3-A antibody that targets the neurotoxic A peptide was recently developed by a researcher. The antibody also contains a component that hinders C1q binding and complement system activation, reducing microglial activation in Alzheimer’s patients [76]. In another study, mice with amyotrophic lateral sclerosis were administered the C5aR (C5a receptor) antagonist PMX205, which was successful in penetrating the central nervous system, enhancing grip strength, and slowing disease progression in the hind limbs [77].

### 4.2. Age-Related Macular Degeneration

Age-related macular degeneration (AMD) is the leading cause of blindness in elderly populations in Western countries, and both age and genetics are risk factors for AMD [78]. AMD is characterized by the early appearance of extracellular vitreous deposits between the choroid and the retinal pigment epithelium, which trigger inflammation. Late onset choroidal neovascularization (CNV) and retinal pigment epithelial (RPE) cell death are linked to severe vision impairment [79]. One of the main causes of AMD is overactivation of the complement replacement pathway [80]. The C3d/C3 ratio was shown to be considerably higher in AMD patients than in controls, and complement activation was higher during the AMD disease phase [81]. C3a can also cause inflammation by stimulating mast cell degranulation, which can facilitate AMD development [82,83]. Moreover, in the early stages of AMD, C3a and C5a are deposited in the subretinal pigment epithelium, increasing the production of inflammatory proteins. Knockout of the C3a and C5a receptor genes reduced vascular endothelial growth factor (VEGF) expression and CNV following laser injury, and the same effect could be obtained using antibody-mediated inhibition of C3a or C5a receptors [84]. Another study showed that animals injected with adenovirus-based C3 in the subretinal cavity showed a variety of AMD symptoms, including endothelial cell proliferation and migration, or retinal pigment epithelium atrophy, all of which point to a clear link between complement levels and AMD [85,86]. The same effects occurred in a study that used C3^−/−^ mice and WT mice to model AMD, and the C3^−/−^ animals had a more severe response [87], showing that a total loss of C3 was harmful to retinal health and that the balance of complement levels inside the organism is critical. In a phase II clinical trial, the C3 inhibitor APL-2 showed promise in slowing the progression of GA (gyrate atrophy of the choroid and retina). The C5 inhibitor avacincaptad pegol showed preliminary efficacy against AMD in a phase 2 trial [88].

### 4.3. Osteoarthritis

Osteoarthritis (OA) is the most common kind of arthritis and is a chronic aging disease that is one of the primary causes of mobility loss in middle-aged and older individuals [89]. Mechanical and biological factors contribute to joint degradation, and metabolic and endocrine diseases lead to distinct phases of osteoarthritis. The deterioration of articular cartilage and subsequent destruction of the subchondral bone are the major hallmarks of OA [90]. Evidence that the complement system is linked to osteoarthritis is emerging. First, C4d and C3bBbP levels are 2–34 times higher in OA, rheumatoid arthritis (RA), and pyrophosphate arthritis (PPA) patients than in controls [91]. C3, C4, and Fb have been found in various tissues in osteoarthritis patient joints, and in vitro cultures have shown that chondrocytes can produce C3a and C5a [92]. In osteoarthritis model mice, the MAC is required for the development of inflammation, and chondrocytes from C5-deficient animals exhibited less inflammation than controls. Similarly, through metabolic reprogramming of synovial fibroblasts, C3 and C3a can stimulate the development of local inflammation in tissue and knocking down C3 can minimize osteoarthritic regions and reduce inflammatory damage [93,94]. Because C3a and C5a are linked to post inflammatory pain, C3aR and C5aR could be used to treat chronic inflammatory pain [95,96]. However, C3a and C5 also play beneficial roles. These factors attract mesenchymal stem cells (MSCs), which can help with OA bone repair and stimulate joint angiogenesis. As a result, the balance of complement levels in the organism is critical, and fully eliminating C3 or C5 may not be the optimum strategy. PMX53, a C5aR antagonist, was first used in clinical trials to treat RA, OA, and AMD, and it was found to be safe in both phase I and phase II trials. However, because it was poorly tolerated when taken orally, it was replaced with a newer form, PMX205, which will be used in combination with other drugs to treat these diseases in the future [97,98,99,100].

### 4.4. Cardiovascular Disease

Aging leads to insulin resistance and dyslipidemia through induction of low-grade inflammation and inhibition of lipogenic differentiation eventually leading to the development of cardiometabolic diseases, of which cardiovascular disease (CVD) is the main clinical manifestation, characterized by severe stenosis or occlusion of blood vessels forming thrombi [101,102]. It has been shown that damaged vascular endothelial cells can serve as a physiological source of plasma properdin. Preprotein is a positive regulator of complement activation and aggravates the activation of complement [103]. Complement is a valuable biomarker for cardiovascular disease, and the researchers found through 78 patients with cardiovascular disease that C3a was elevated in all 43 patients with unstable angina and was not significantly elevated in 35 patients with stable angina [104]. Patients with congestive heart failure were shown to have higher amounts of C3 split-products as well as sC5b-9 (soluble MAC complex) [105]. CRP was discovered to be deposited solely in the infarcted areas of the heart, not in the healthy cardiac tissue, in patients who died from AMI (Acute Myocardial Infarction) [106]. Furthermore, the deposition of CRP occurred simultaneously with the deposition of C4 and C3. A study including individuals with acute coronary syndromes (ACS), a condition of ongoing inflammation, found elevated C3 and C4 levels [107]. Middle-aged males with high C3 serum levels were found to be at an increased risk for myocardial infarction [108], while high C3 levels in women were linked to atherosclerotic problems [109].

### 4.5. Other Diseases

Complement is not only related to the above diseases, but also to disease of the renal disease, lung disease, and liver disease. Complement system is implicated in various aspects of renal disease; a function for complement in renal disease was inferred from the deposition of complement components in afflicted kidneys and variations in complement levels in the circulation during active disease [110]. C1q, C4, and C2 deficiency significantly increases the risk of developing systemic lupus erythematosus (SLE) through mechanisms including faulty apoptotic material clearance. Lupus nephritis, which is frequently seen in SLE patients, is characterized by the accumulation of immune complexes throughout the glomerulus and large infiltrates [110,111]. In addition, in many pulmonary diseases, complement may act as a crucial connection between innate and adaptive immunity [112]. The injection of soluble CR1, a C3 and C5 convertase inhibitor, increased oxygenation of the transplanted lung and reduced pulmonary edema in a swine model [113,114]. Keshavjee et al. demonstrated that CR-1 inhibition given to human lung transplant recipients reduced the duration of mechanical ventilation and enhanced gas exchange [115]. An orthotopic allogeneic single left lung transplantation model in rats showed reduced reperfusion injury and enhanced oxygenation when the therapeutic effects of inhibiting complement and leukocyte adhesion molecules were combined [112,116]. Complement has also been involved in liver regeneration after partial hepatectomy or after toxic injury [117,118]. Strey et al. showed that the anaphylatoxins C3a and C5a are crucial for liver regeneration using a mouse model of partial hepatectomy [119]. C3 or C5 deficiency causes impaired liver regeneration, as well as temporary or catastrophic liver failure following partial hepatectomy. In C3 and C5 double deletion mice, liver regeneration was substantially compromised, but it was recovered after simultaneous reconstitution with C3a and C5a [119,120].

In short, although the roles of complement in different diseases are still different, a large number of studies have shown that over-activation of complement significantly induces disease and balancing with complement is conductive to disease prevention and treatment. More studies examining the role of complement in diseases in the human body are needed.

## 5. Conclusions

Aging is a complicated process that is accompanied by a variety of chronic diseases as various bodily functions gradually deteriorate. We cannot avoid aging, but we can ameliorate damaging effect. The complement system, which is part of the innate immune system, is responsible for not only pathogen elimination but also immunomodulation. Furthermore, our research shows that complement composition is linked to aging, suggesting that complement composition could be used to monitor aging or health. The complement system is a potential target for therapeutic intervention in aging or aging-related diseases. However, when using complement as a therapeutic target, we should pay attention to the balance of complement; too much can exacerbate the disease, while too little can be counterproductive, which is critical for aging-related disease diagnosis and precise treatment.

## Figures and Tables

**Figure 1 ijms-23-08689-f001:**
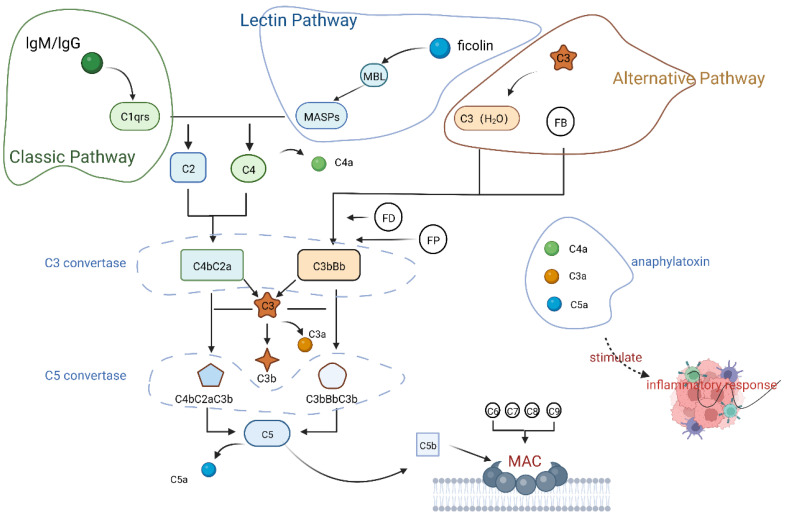
Complement system activation pathways. Created with BioRender.com (accessed on 13 June 2022).

## Data Availability

Not applicable.

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
