# Peer review of "The Complement System, Aging, and Aging-Related Diseases"

_ijms, 2022, doi:10.3390/ijms23158689_

Round 1

Reviewer 1 Report

The review entitled ‘Complement system, Aging and Aging-related Diseases’ by Zheng et al, provides an overview of the effect of complement system on aging and aging-related diseases, such as Alzheimer's disease, age-related macular degeneration, and osteoarthritis. It is a relevant theme that needs more attention and discussion, making possible a better understanding of the complex aging process and new therapeutic approaches for aging-related diseases. The authors brought a brief description of the complement system, its function, composition and activation pathways, followed by a description of the role of complement in aging, inflammation, metabolism, apoptosis and the aging-related diseases.

Overall, the manuscript presents relevant and recent references but requires some English adjustments and minor revisions listed below.  

Minor revisions:

Some complement components, such as C3d and C4d, are mentioned in the text without previous information of what they are or what they are product of. Plus, some important terms are abbreviated in the text without previous extense written. It should be taken attention to that.

Line 33: “complement” should be “complement system” as it is the first mention of it in Introduction section.

Line 57: this article summary” should be “this article summarizes”.

Line 65: “heat-insensitive” should be “heat-sensitive”, as the complement system is sensitive to high temperatures.

Lines 75: “AP” should be alternative pathway, since it is the first mention in the text. 

Lines 77: C5 convertase does not “recruits MACs”. It cleaves the C5 component and its activated form, C5b, binds to cell surface and triggers MAC assembly. The correction should be done.

Line 81/82: “class pathway” should be “classical pathway”.

Line 111: “Alternate pathway” should be “alternative pathway”.

Lines 119-121: The sense of the phrase is not clear. It should be rewritten in a clearer form.

Lines 128-129: The sense of the phrase is not clear. It should be rewritten in a clearer form.

Line 132: Properdin is the important positive stabilizer of the AP C3 convertase and should be mention in the AP description.

Line 140: “AP and LP” should be “CP and LP”.

Line 166: C4A should be C4.

Line 175: “C3 and inflammatory” should be C3 and inflammation.

Line 300: “C4d,C3bBbP levels” should be C4d and C3bBbP levels.

Line 311: The sentence “C3a and C3a and C5a are also nice.” is not making sense. It should be written in a clearer form.

Figure 1: In the Lectin pathway, “MLP” should be “MBL”. And Properdin should be included in the Alternative pathway C3 convertase.

Figure 2: The meanings of the abbreviatons should be in the legend of the figure for the sake of clarity.

Reviewer 2 Report

In this review, authors aimed to compile recent information related to the role of the complement system in age-related diseases. However, during the manuscript the focus of the review seems to be diffuse and not well defined. Furthermore, multiple reviews about the complement and aging have been published during the last 4 years. There is not additional information or a novel interpretation of existing literature, compared with published reviews. Importantly, the text is not well written using the scientific language.

Round 2

Reviewer 2 Report

Authors have edited the manuscript in order to use a style more accordingly to a scientific review. However, the content of the manuscript has not improved importantly. It would be interesting to read studies performed in aging models and to learn about the molecular mechanisms behind the connection between complement and aging. 

Author Response

Dear Reviewers:
